# Robust Visual-Inertial Integrated Navigation System Aided by Online Sensor Model Adaption for Autonomous Ground Vehicles in Urban Areas

**Xiwei Bai [1], Weisong Wen [2] and Li-Ta Hsu [1,*]**

1   Interdisciplinary Division of Aeronautical and Aviation Engineering, The Hong Kong Polytechnic University, Kowloon, Hong Kong 999077, China; xiwei.bai@connect.polyu.hk
2   Department of Mechanical Engineering, The Hong Kong Polytechnic University, Kowloon, Hong Kong 999077, China; weisong.wen@connect.polyu.hk
*   Correspondence: lt.hsu@polyu.edu.hk

**Abstract:** The visual-inertial integrated navigation system (VINS) has been extensively studied over the past decades to provide accurate and low-cost positioning solutions for autonomous systems. Satisfactory performance can be obtained in an ideal scenario with sufficient and static environment features. However, there are usually numerous dynamic objects in deep urban areas, and these moving objects can severely distort the feature-tracking process which is critical to the feature-based VINS. One well-known method that mitigates the effects of dynamic objects is to detect vehicles using deep neural networks and remove the features belonging to surrounding vehicles. However, excessive feature exclusion can severely distort the geometry of feature distribution, leading to limited visual measurements. Instead of directly eliminating the features from dynamic objects, this study proposes to adopt the visual measurement model based on the quality of feature tracking to improve the performance of the VINS. First, a self-tuning covariance estimation approach is proposed to model the uncertainty of each feature measurement by integrating two parts: (1) the geometry of feature distribution (GFD); (2) the quality of feature tracking. Second, an adaptive M-estimator is proposed to correct the measurement residual model to further mitigate the effects of outlier measurements, like the dynamic features. Different from the conventional M-estimator, the proposed method effectively alleviates the reliance on the excessive parameterization of the M-estimator. Experiments were conducted in typical urban areas of Hong Kong with numerous dynamic objects. The results show that the proposed method could effectively mitigate the effects of dynamic objects and improved accuracy of the VINS is obtained when compared with the conventional VINS method.

**Keywords:** Visual-inertial integrated navigation system (VINS); dynamic objects; adaptive tuning; positioning, autonomous systems; urban canyons

## 1. Introduction

In recent years, the visual-inertial integrated navigation system (VINS) has had important applications in various fields due to its cost-efficiency, for example, unmanned aerial vehicles (UAV) [1,2], augmented reality (AR) [3] and autonomous ground vehicle (AGV) positioning [4–6]. There were significant achievements from research on the VINS, such as the VINS-Mono [6], visual-inertial direct sparse odometry (VI-DSO) [7] and semi-direct visual odometry (SVO) [8]. These existing methods have good performances in an ideal environment with sufficient texture information and static environmental features. In other words, the VINS relies heavily on the assumption that the surrounding features are static. However, the performance of the VINS can be significantly impaired

in dynamic outdoor scenarios as motion blur of images damages the quality of features tracking [9]. As shown in Figure 1, there are numerous dynamic objects, such as vehicles and pedestrians, in a typical urban scenario. As a result, pose estimation from the VINS can significantly drift or even fail due to degraded feature tracking caused by dynamic objects [10], such as moving vehicles and pedestrians. Our previous study in [11] evaluated the performance of a state-of-the-art VINS method, the VINS-Mono [6], in diverse urban canyons with numerous dynamic objects. The results show [11] that dynamic objects are one of the major reasons why the performance of the VINS is degraded in urban areas. Major directions of researches on mitigating the effects of dynamic objects on the accuracy of the VINS include: (1) dynamic object detection [12] based on motion tracking; (2) moving object detection and removal based on deep learning [13]; (3) mitigating the effects of dynamic objects using robust methods.

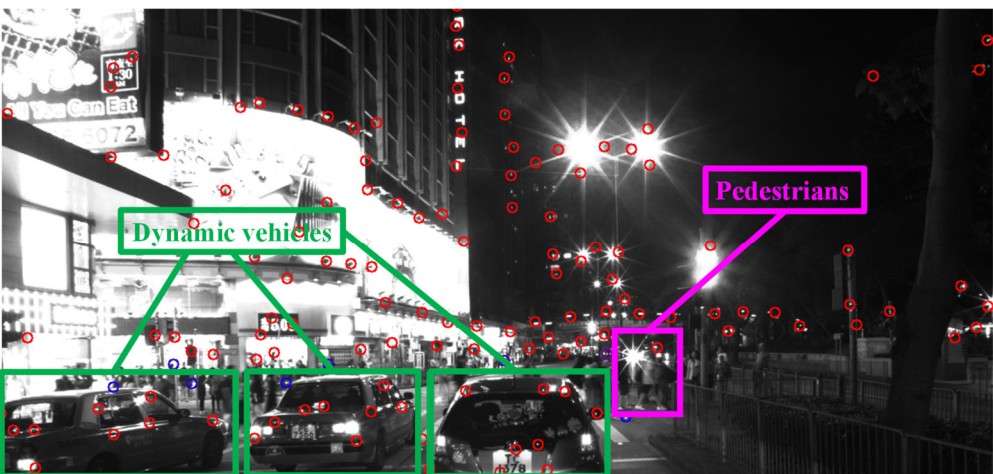

**Figure 1.** Illustration of a typical urban scenario with numerous dynamic objects, such as dynamic vehicles and pedestrians.

The motion tracking-based methods [14,15] are proposed to mitigate the effects of the dynamic objects by detecting and remodel their features belonging to dynamic objects. Generally, the principle is to identify the features or pixels that are associated with moving objects. A pixel-wise segmentation motion approach was introduced in [15], which proposed an online RGB-D data-based motion removal method. It proposes to filter out data related to moving objects. However, one of the major limitations is that large parallax can degrade the performance of foreground segmentation and cause motion tracking failure. Similar researches are conducted in [16–18] where an RGB-D camera was used for dynamic object detection and tracking. However, the maximum ranging of the RGB-D camera is limited (usually between 8 and 10 meters [19]) which is not satisfactory for outdoor applications such as UAVs and high-speed AGVs. Moreover, the motion tracking-based method to detect the dynamic object relies heavily on the accuracy of vehicle egomotion estimation [20,21], which is a major challenge.

The straightforward method to mitigate the effects of dynamic objects is to detect and remove the features of the dynamic objects from visual simultaneous localization and mapping (SLAM) [22,23]. Due to the excessive dynamic objects in complex environments, a detect-SLAM system [24] is proposed to integrate SLAM with a deep neural network to detect moving objects and remove the unreliable features from moving objects. The DynaSLAM system [25] introduces the convolutional neural network (CNN) to segment the images so that features belonging to the dynamic objects are rejected. Alternatively, an SSD detector [26] is presented to detect moving objects with a priori knowledge and a selection tracking algorithm is proposed to remove dynamic objects. In addition, an ML-RANSAC algorithm [23] was proposed to distinguish moving objects from stationary objects and classify the outliers belonging to moving objects. Although numerous researches were conducted on object detection [27–29], there are still many challenges for dynamic object detection. Many object detection systems based on deep learning, such as the state-of-the-art YOLO [29] and FPN [30], can

detect objects (vehicles, trucks and pedestrians), but cannot determine the movement status of these objects (static or dynamic). More important, these existing methods tend to remove the features from dynamic objects. However, the performance of the VINS relies heavily on the number of features [10] and the geometry of feature distribution (GFD) [31]. Excessive exclusion of dynamic feature points (DFPs) can severely degrade the quality of the feature tracking process. Therefore, removing all the DFPs is not acceptable in urban canyons.

Instead of directly removing all the detected DFPs from visual SLAM, adaptively estimating the covariance of visual measurements to further de-weight the effects of DFPs for visual SLAM attracts lots of attention [32,33]. The adaptive covariance estimation was proposed in [32] to enhance the resilience against dynamic objects in cooperative visual SLAM. However, instant information communication with low latency is required which is usually not available for commercial level applications. Recently, the state-of-the-art method, the switchable constraints [34] was proposed, which can probabilistically detect and de-weight the outlier measurements from factor graph optimization (FGO) and improved performance was obtained. However, it relies heavily on the accuracy of the initial guess of prior switchable constraints [34]. Moreover, it requires redundancy of healthy measurements. In other words, the switchable constraints can deliver decent performance only when the number of healthy measurements significantly exceeds the outlier measurements. In addition, each feature can derive a switchable constraint factor in FGO which can cause an unacceptable computational load in the VINS subsequently. Recently, the dynamic covariance estimation (DCE) [35] algorithm was proposed to mitigate the effects of GNSS (global navigation satellite systems) outlier measurements and significantly improved accuracy is obtained with real-time performance. The uncertainty of GNSS measurements and the state are estimated simultaneously. However, the method relies heavily on the initial guess about the states to calculate reliable residuals [35]. Similar work was done in [36]. Moreover, the M-estimator algorithm [37] is applied to further enhance robustness against GNSS outliers in [36]. The principle of the M-estimator in FGO is to embed an additional robust function in the standard error function, such as Cauchy [38] and Huber [38] functions. However, the performance of the applied M-estimator relies heavily on tuning its parameters. In other words, the parameters of the M-estimator have to be carefully tuned based on the scenarios to obtain expected performance. Similarly, the M-estimator is also used to resist the outlier measurements in the VINS. In [6], the tightly coupled integration of the visual-inertial system is designed for state estimation of autonomous drones and an M-estimator was used to increase the robustness of the VINS. However, the improvement of the performance of the VINS through the M-estimator is limited in dynamic scenarios. Similar research was extended in [39,40] and the same framework was used. The M-estimator was applied to increase the robustness of the standard error function and improved performance was obtained. However, the performance of the M-estimator is still limited by parameter tuning.

In fact, the principle of visual and global navigation satellite systems (GNSS) is similar in positioning, both requiring referenced positioning from visual measurements or receptions of satellites. The references for visual-based positioning are the tracked features and the ones for GNSS positioning are the pseudorange measurements from satellites, which can be seen in Figure 2. The major difference is that the VINS requires abundant feature tracking as the pose of features is unknown. However, the GNSS only requires a minimum of five satellites to achieve the positioning of the GNSS receiver and the positions of satellites are known. Interestingly, similar positioning problems can also be seen in the GNSS which is based on signals received from multiple satellites [41]. The non-line-of-sight (NLOS) receptions are similar to the dynamic feature points (DFPs) in the VINS as both are the unhealthy (outlier) measurements. As shown in Figure 2, the satellite is blocked by the building, leading to the NLOS (red satellites) receptions which are similar to the DFP (the red one on the right side of Figure 2). Excluding all NLOS satellites will severely distort the geometry distribution of satellites in deep urban areas and even cause a lack of satellites for further positioning. In our latest research [42] on GNSS positioning, both the NLOS and line-of-sight (LOS) measurements are utilized by giving them different weightings and improved positioning performance was obtained. Therefore, we believe that remodeling outlier measurement is preferable. Interestingly, our previous

work in [11] extensively evaluated the performance of the VINS in urban canyons and we find out that the positioning error is closely related to the quality of feature tracking with almost a linear relationship [11]. Moreover, our recent work [43] shows that the excessive exclusion of DFPs can distort the geometry of feature distribution, which can degrade the performance of the VINS. Inspired by the work in [35,36] and our findings in [11,43], this study proposes to estimate the sensor model of visual measurements (tracked feature) online based on the quality of feature tracking. First, a self-tuning covariance estimation approach is proposed to model the uncertainty of the feature measurement by integrating two parts: (1) the geometry of feature distribution (GFD); (2) the quality of feature tracking. Second, an adaptive M-estimator is proposed to correct the measurement residual model to further mitigate the effects of the outlier measurements, such as the dynamic features. Unlike conventional M-estimator, the proposed method effectively relax the reliance on the excessive parameterization of the M-estimator.

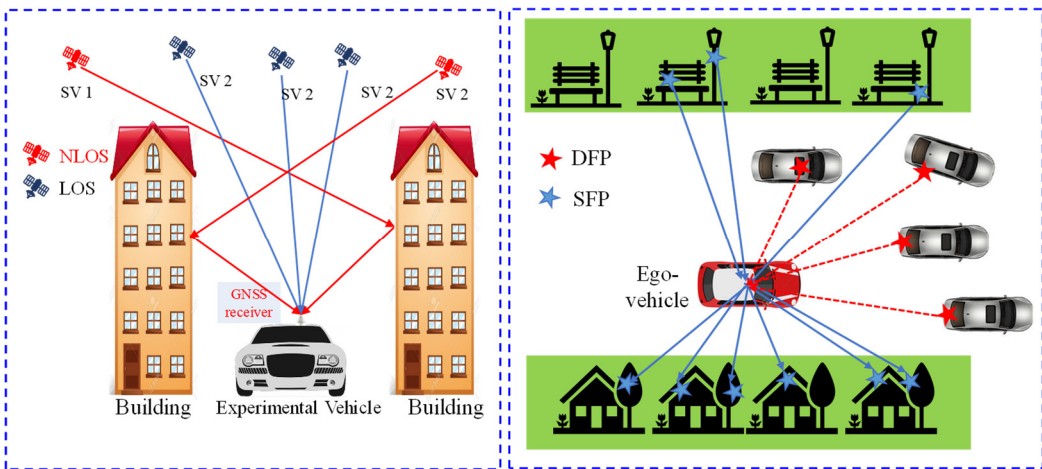

**Figure 2.** The positioning principles of global navigation satellite systems (GNSS) and visual odometry. LOS denotes line of satellite. SFP denotes static feature point and DFP denotes dynamic feature point.

The rest of this study is organized as follows: In Section 2, an overview of the proposed method is given; then the framework of the VINS is described in Section 3. Next, the online sensor model estimation is presented in Section 4. The experiment result is shown in Section 5. Finally, a conclusion is drawn.

## 2. Overview of the Proposed Adaptive VINS

The structure of the proposed method is shown in Figure 3. The inputs of the system consist of two parts: the inertial measurement unit (IMU) and the monocular camera. The IMU provides acceleration and angular velocity measurements at a high frequency. The monocular camera provides raw images. In the modeling stage, due to the high data frequency of the IMU, multiple IMU measurements are obtained between two consecutive frames. To reduce the computational loads, the IMU pre-integration [44] is employed to derive the motion between the consecutive frames. Then the pre-integration factor is obtained. In addition, the features are extracted and tracking for visual modeling is performed. To adaptively estimate the uncertainty of feature measurements, two parameters are considered in this study. The geometry of feature distribution is derived from feature extraction. The number of times of the features being tracked is derived from the feature tracking. Both are used in the proposed adaptive covariance estimation, which can remodel the uncertainty of visual measurements. Then the standard reprojection factor is obtained based on the adaptive covariance estimation. In addition, the quality of feature tracking is associated with the parameters of the adaptive M-estimator, which can increase the robustness of the standard reprojection factor against outlier measurements with the additional robust function. Then the robust reprojection factor is obtained. Finally, the pre-integration factor and robust reprojection factor are integrated into an

FGO. The optimization result can correct the bias of IMU measurements in turn. In short, the contributions of this study are listed as follows:

(1) This study proposes a self-tuning covariance estimation approach to model the uncertainty of each feature measurement by integrating two parts: (1) the geometry of feature distribution (GFD); (2) the quality of feature tracking;

(2) This study proposes an adaptive M-estimator to correct the measurement residual model to further mitigate the effects of outlier measurements, like the dynamic features. The proposed adaptive M-estimator effectively relaxed the drawback of manual parameterization [36] of M-estimator;

(3) This study employs challenging datasets collected in dynamic urban canyons of Hong Kong to validate the effectiveness of the proposed method in mitigating the effects of dynamic objects. improved performance is achieved compared with the state-of-the-art VINS solution [6].

The details about the proposed method are given in the following sections.

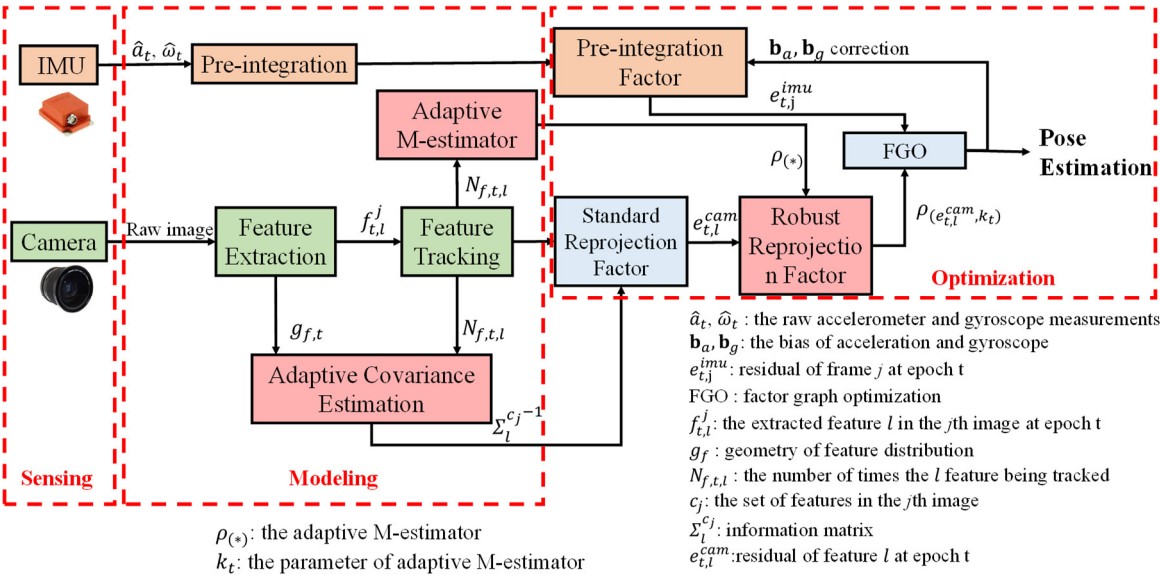

**Figure 3.** Flowchart of the proposed visual-inertial integrated navigation system (VINS) framework aided by adaptive covariance estimation and M-estimator.

## 3. Tightly Coupled Monocular-based Visual-inertial Integration based on Factor Graph Optimization

### 3.1. System States

The objective of FGO is to minimize the residuals derived from multiple sensor measurements [45]. In this study, the residuals include the one from the IMU measurements and the one from visual measurements. The state vector considered in this study is defined as follows:

$$\chi = [x_1, x_2, \dots, x_n, x_c^b, \lambda_1, \lambda_2, \dots \lambda_M] \tag{1}$$

$$x_k = [P_{b_k}^w, V_{b_k}^w, q_{b_k}^w, b_{a,k}, b_{g,k}], \ k\epsilon[1,n] \tag{2}$$

$$x_c^b = [P_c^b, q_c^b] \tag{3}$$

where the superscript $w$ is the world frame and the subscript $b_k$ is the body frame (same as the IMU frame) while the $k$th image is captured. $\mathbf{x}_k$ is the IMU state at the $k$th image. It contains the position ($\mathbf{P}_{b_k}^w$), the velocity ($\mathbf{V}_{b_k}^w$) and the orientation that is represented by quaternion ($\mathbf{q}_{b_k}^w$) in the world frame and the acceleration bias ($\mathbf{b}_{a,k}$) and the gyroscope bias ($\mathbf{b}_{g,k}$) in the IMU body frame. $n$ is the total number of keyframes utilized for optimization and $M$ is the total number of features considered. $\lambda_l$ is the inverse depth of the $l$th feature observed for the first time, $l \in (1, M)$. $\mathbf{x}_c^b$ is the extrinsic parameter that transforms the camera frame into the IMU frame. To guarantee the computation efficiency, we only utilize the measurements inside a sliding window (which can be seen in Figure 4) to estimate the states. The images inside in the sliding window are between the frames $b_k$ and $b_{k+n}$, with the time of $t_k$ and $t_{k+n}$, respectively. Regarding the implementation of the VINS, we refer to the framework proposed in [6].

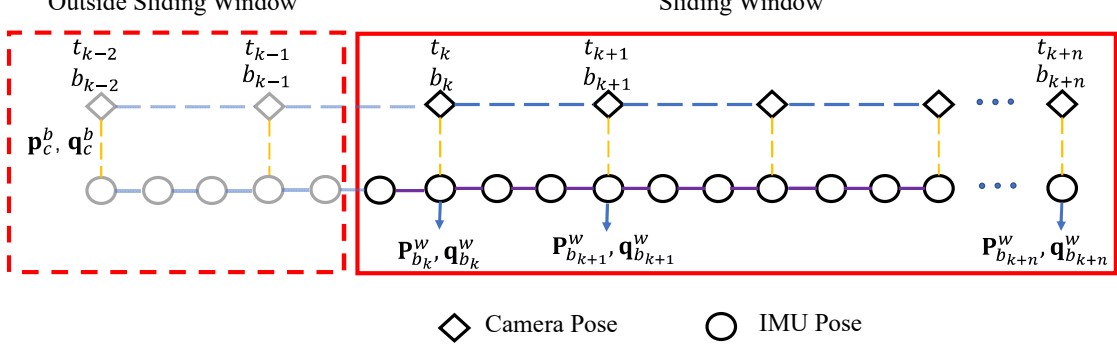

**Figure 4.** Illustration of the sliding window used in the proposed graph optimization. The poses inside the red and dash rectangle denote the marginalized states. The poses inside the red and solid rectangle represent the states considered in factor graph optimization (FGO).

### 3.2. IMU Measurement Modeling

This section presents the modeling of IMU measurements. The IMU measurements are given in the body frame, which is affected by the additive noise and bias of acceleration and gyroscope. The raw accelerometer and gyroscope measurements at a given time $t$ are expressed as follows:

$$\hat{a}_t = a_t + R_w^t g^w + b_{a_t} + n_a \tag{4}$$

$$\hat{\omega}_t = \omega_t + b_{\omega_t} + n_\omega \tag{5}$$

where $\hat{a}_t$ and $\hat{\omega}_t$ denote the raw measurements of the IMU, $a_t$ and $\omega_t$ are the expected measurements of acceleration and angular velocity, $g^w = [0 \quad 0 \quad g]^T$ denotes the gravity vector in the world frame, $R_w^t$ denotes the rotation matrix that encodes the transformation the world frame into the body frame at time $t$, $b_{a_t}$ and $b_{\omega_t}$ denote the acceleration bias and gyroscope bias and $n_a$ and $n_\omega$ are the additive noise, which is assumed to be Gaussian white noise, $n_a \sim \mathcal{N}(0, \sigma_a^2)$ and $n_\omega \sim \mathcal{N}(0, \sigma_\omega^2)$. The values of $n_a$ and $n_\omega$ are determined based on the specifications of the applied IMU sensor.

The IMU measurements can be employed to constrain the motion between two epochs using the standard IMU mechanism [46], which can work efficiently in the filtering-based sensor fusion, such as the extended Kalman filter (EKF) [47]. However, the standard IMU mechanism [46] can cause a high computation load in sensor fusion using FGO, due to the high frequency of the IMU measurements. Therefore, we employ the state-of-the-art IMU pre-integration technique [44] to integrate the IMU measurements, which can significantly alleviate the high computation load in FGO and the accuracy is guaranteed, by integrating multiple IMU measurements into a single factor in FGO. There are several inertial measurements in the time interval $t \in [t_k, t_{k+1}]$ between the two consecutive frames $b_k$ and $b_{k+1}$. With the given bias estimation, the IMU pre-integration is integrated in the $b_k$ frame as follows [6]:

$$\alpha_{b_{k+1}}^{b_k} = \iint_{t\in[t_k,t_{k+1}]} R_t^{b_k}(\hat{a}_t - b_{a_t})dt^2 \tag{6}$$

$$\beta_{b_{k+1}}^{b_k} = \int_{t\in[t_k,t_{k+1}]} R_t^{b_k}(\hat{a}_t - b_{a_t})dt \tag{7}$$

$$\gamma_{b_{k+1}}^{b_k} = \int_{t\in[t_k,t_{k+1}]} \frac{1}{2}\Omega(\hat{\omega}_t - b_{\omega_t})\gamma_t^{b_k}dt \tag{8}$$

$$\Omega(\omega) = \begin{bmatrix} 0 & -\omega_z & \omega_y & \omega_x \\ \omega_z & 0 & -\omega_x & \omega_y \\ -\omega_y & \omega_x & 0 & \omega_z \\ \omega_x & \omega_y & \omega_z & 0 \end{bmatrix} \tag{9}$$

where $\alpha_{b_{k+1}}^{b_k}$, $\beta_{b_{k+1}}^{b_k}$ and $\gamma_{b_{k+1}}^{b_k}$ are the pre-integration terms between the frames $b_k$ and $b_{k+1}$, which represent the changes of position, velocity and orientation, respectively. $R_t^{b_k}$ is the rotation matrix that transforms the body frame at time $t$ into the reference frame $b_k$. In fact, this is one of the major differences from the IMU mechanism, as the pre-integration is performed in the body frame $b_k$ and the IMU mechanism is conducted concerning the world frame. $\gamma_t^{b_k}$ is a quaternion that transforms the body frame at time $t$ into the reference frame $b_k$. $\omega_x$, $\omega_y$ and $\omega_z$ denote the angular velocities in the body frame.

The IMU pre-integration between the two consecutive frames uses $b_k$ as the reference frame. Based on the information, the position, the velocity and the orientation in the world frame can be derived as follows:

$$P_{b_{k+1}}^w = \left(P_{b_k}^w + V_{b_k}^w \Delta t_k - \frac{1}{2}g^w \Delta t_k^2\right) + R_{b_k}^w \alpha_{b_{k+1}}^{b_k} \tag{10}$$

$$V_{b_{k+1}}^w = \left(V_{b_k}^w - g^w \Delta t_k\right) + R_{b_k}^w \beta_{b_{k+1}}^{b_k} \tag{11}$$

$$\gamma_{b_{k+1}}^{b_k} = q_w^{b_k} \otimes q_{b_{k+1}}^w \tag{12}$$

The symbol $\otimes$ means multiplication between two quaternions. According to the two states $(\mathbf{P}_{b_{k+1}}^w$ and $\mathbf{P}_{b_k}^w)$ of $b_k$ and $b_{k+1}$, the residual for IMU pre-integration measurements in the two consecutive frames $b_k$ and $b_{k+1}$ can be defined as follows [6]:

$$r_{\mathcal{B}}\left(\hat{Z}_{b_{k+1}}^{b_k},\chi\right) = \begin{bmatrix} \delta\alpha_{b_{k+1}}^{b_k} \\ \delta\beta_{b_{k+1}}^{b_k} \\ \delta\theta_{b_{k+1}}^{b_k} \\ \delta b_a \\ \delta b_\omega \end{bmatrix} = \begin{bmatrix} R_w^{b_k}\left(P_{b_{k+1}}^w - P_{b_k}^w + \frac{1}{2}g^w \Delta t_k^2 - V_{b_k}^w \Delta t_k\right) - \alpha_{b_{k+1}}^{b_k} \\ R_w^{b_k}\left(V_{b_{k+1}}^w + g^w \Delta t_k - V_{b_k}^w\right) - \beta_{b_{k+1}}^{b_k} \\ 2\left[q_{b_k}^{w^{-1}} \otimes q_{b_{k+1}}^w \otimes (\gamma_{b_{k+1}}^{b_k})^{-1}\right]_{xyz} \\ b_{a,b_{k+1}} - b_{a,b_k} \\ b_{\omega,b_{k+1}} - b_{\omega,b_k} \end{bmatrix} \tag{13}$$

Where the variable $\hat{Z}_{b_{k+1}}^{b_k}$ represents the observation measurements of the IMU between the frames $b_k$ and $b_{k+1}$. The operator $[.]_{xyz}$ is used for extracting the vector part of the quaternion q for the orientation difference. $\Delta\theta_{b_{k+1}}^{b_k}$ represents the orientation constraint between the frames $b_k$ and $b_{k+1}$. $\delta\alpha_{b_{k+1}}^{b_k}$ represents the derived position constraint between the frames $b_k$ and $b_{k+1}$. $\delta\beta_{b_{k+1}}^{b_k}$ denotes the velocity constraint. $\delta b_a$ and $\delta b_\omega$ denote the accelerometer and gyroscope biases constraints, respectively. $\left[\alpha_{b_{k+1}}^{b_k}, \beta_{b_{k+1}}^{b_k}, \gamma_{b_{k+1}}^{b_k}\right]$ represents the pre-integration measurements between

the frames $b_k$ and $b_{k+1}$. When the estimation of bias changes, the IMU measurements will be repropagated based on the new bias estimation.

### 3.3. Visual Measurement Modeling

This section presents the modeling of visual measurement. The direct raw measurement from the camera is the raw image at a given epoch $t$. Similar to the work in [6], we formulate the visual measurement residual based on a reprojection error. For a given new image, the features are detected using the Shi–Tomasi [48] corner detection algorithm. Meanwhile, the Kanade–Lucas–Tomasi (KLT) sparse optical flow algorithm [49] is employed to track the features. The derivation of the reprojection error relies heavily on the quality of feature tracking. To guarantee that enough features are detected in a frame of the image, new corner features are also detected [48]. During the feature tracking, only certain images, the keyframes, are employed to perform the feature tracking to improve efficiency. The keyframes are chosen based on two criteria: (1) The first one is the average parallax criteria: if the average parallax of the tracked features between the current frame and the latest keyframe override a certain threshold, the current frame is treated as a new keyframe. (2) if the number of tracked features inside the current image is smaller than a certain threshold, this frame is regarded as a new keyframe. Figure 5 shows the feature tracking process where $n$ denotes the total number of keyframes inside the sliding window. The $l$th feature is first, observed in the $i$th image. $Z_l^{c_i}(\hat{u}_l^{c_i}, \hat{v}_l^{c_i})$ represents first observation of the $l$th feature in the $i$th image. $Z_l^{c_j}(\hat{u}_l^{c_j}, \hat{v}_l^{c_j})$ denotes the observation of the same feature in the $j$th image. We can see from Figure 5 that the feature is tracked for several times. In other words, the features are seen in several image fames.

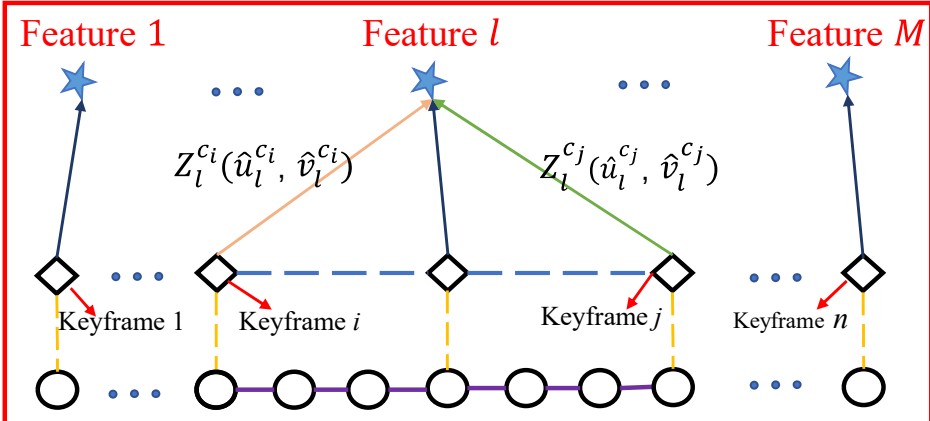

$M$ : total number of features in the sliding window
$c_i$, $c_j$ : the set of features observed in the $i$th and $j$th images

$Z_l^{c_i}$, $Z_l^{c_j}$ : observation of the $l$th feature in the $i$th and $j$th images
$(\hat{u}_l^{c_i}, \hat{v}_l^{c_i})$, $(\hat{u}_l^{c_j}, \hat{v}_l^{c_j})$ : the pixel location of $l$th feature in $i$th and $j$th images

**Figure 5.** Illustration of the feature-tracking process.

The traditional reprojection residual is defined in the image plane, which is not suitable for most camera models. In this study, following the work in [6], the residual is defined on a unit sphere, which is applicable for almost all the camera models. The unit vector for observation of the $l$th feature in the $j$th image that is projected into the unit sphere, $\hat{\bar{p}}_l^{c_j}$, is calculated as follows [6]:

$$\hat{\bar{p}}_l^{c_j} = [I_1 \quad I_2]^T \cdot \pi_c^{-1}\left(\begin{bmatrix} \hat{u}_l^{c_j} \\ \hat{v}_l^{c_j} \end{bmatrix}\right) \tag{14}$$

where $[I_1 \quad I_2]^T$ are two arbitrarily selected orthogonal bases on the tangent plane corresponding to the feature observation. $\pi_c^{-1}$ is the back-projection function, which turns a pixel location into a unit

vector using the camera's intrinsic parameters. To formulate the residual corresponding to the measurement $\hat{\tilde{p}}_l^{c_j}$, the expected observation $p_l^{c_j}$ is needed. The direct method is to derive $p_l^{c_j}$ based on the current state $\chi$. To make full use of the feature-tracking process which provides continuous geometry constraints, we derive $p_l^{c_j}$ based on the keyframe $i$. For the sake of clearer explanation, we divide the formulation into the following steps:

Step 1: obtain feature $l$ from the pixel position in image $i$ to the body frame (IMU frame) as follows:

$$S_1 = R_c^b \frac{1}{\lambda_l} \pi_c^{-1} \left( \begin{bmatrix} \hat{u}_l^{c_i} \\ \hat{v}_l^{c_i} \end{bmatrix} \right) + P_c^b \tag{15}$$

$R_c^b$ and $P_c^b$ represent the rotation matrix and the translation matrix from the camera frame to the body frame. Then the pixel location $(\hat{u}_l^{c_i}, \hat{v}_l^{c_i})$ in the $i$th image is transformed into the body frame.

Step 2: obtain feature $l$ in the $i$th image from the body frame to the world frame and then translate the feature to the $j$th image in the world frame as follows:

$$S_2 = R_{b_i}^w (S_1) + P_{b_i}^w - P_{b_j}^w \tag{16}$$

$R_{b_i}^w$ and $P_{b_i}^w$ are the rotation matrix and the translation matrix which transforms the $l$th feature detected in the $i$th image from the body frame to the world frame. $P_{b_j}^w$ encodes the transformation matrix which transforms the $l$th feature detected in the $j$th image frame from the body frame to the world frame.

Step 3: obtain feature $l$ in the $j$th image from the world frame to the body frame and then transform the feature into the camera frame as follows:

$$S_3 = R_w^{b_j} (S_2) - P_c^b \tag{17}$$

$$p_l^{c_j} = R_b^c (S_3) \tag{18}$$

$R_w^{b_j}$ represents the rotation matrix which transforms the same feature in the $j$th image from the world frame to the $b_j$ frame. $p_l^{c_j}$ denotes the predicted feature measurement on the unit sphere by transforming its first observation in the $i$th image to the $j$th image. $R_b^c$ is the rotation matrix that encodes the transformation from the body frame of IMU to the camera frame. $P_c^b$ is the transformation matrix that transforms the camera frame to the body frame of IMU. The $R_b^c$ and $P_c^b$ compose the extrinsic parameters of camera and IMU.

Step 4: therefore, the residual for the $l$th feature measurement in keyframe $j$ is defined as follows:

$$r_C \left( \hat{Z}_l^{c_j}, \chi \right) = [I_1 \quad I_2]^T \cdot (\hat{\tilde{p}}_l^{c_j} - \frac{p_l^{c_j}}{\|p_l^{c_j}\|}) \tag{19}$$

$r_C(*)$ represents the residual of the $l$th feature measurement in the $j$th image. $\hat{Z}_l^{c_j}$ denotes the observation measurement of the $l$th feature in the $j$th image. Be noted that the degree of freedom of the feature has two dimensions and therefore the residual is projected in the tangent plane. $\hat{\tilde{P}}_l^{c_j}$ denotes the unit vector for the observation of the $l$th feature in the $j$th frame.

### 3.4. Marginalization

Each feature measurement corresponds to a factor in FGO. Therefore, the computational complexity will increase dramatically over time. The straightforward way is to remove part of the old states and their associated measurements. However, this will fail to make use of historical data. To reduce the computational loads and guarantee the accuracy, the marginalization is used. The process of marginalization is to marginalize some older visual measurements. During the system optimization, some of the unsatisfactory IMU states and features are marginalized from the sliding

window into a prior factor. Two strategies were proposed [6] to select marginalized measurements. Firstly, if the second latest frame is a keyframe, it will be kept in the sliding window and meanwhile, the oldest frame is marginalized out with its corresponding measurements. Conversely, if the second latest frame is a non-keyframe, the visual measurements will be left out and the IMU measurements that are connected to this non-keyframe are kept, which can maintain the sparsity of the system. The marginalization is carried out by the Schur complement [50]. A new prior is constructed based on all marginalized measurements related to the removed state and the residual for the prior factor can be derived accordingly.

*3.5. Visual-Inertia Optimization*

The objective of the FGO is to minimize the sum of prior and the Mahalanobis norm of all measurement residuals to obtain a maximum posterior estimation. Based on the derived residuals from three parts: (1) residual from IMU pre-integration, (2) residual from the visual measurement and (3) residual from marginalization, the objective function of the system can be formulated as follows:

$$\min_{\chi}\left\{\|r_p - H_p\chi\|^2 + \sum_{k\in\mathcal{B}}\left\|r_{\mathcal{B}}\left(\hat{z}_{b_{k+1}}^{b_k},\chi\right)\right\|^2_{P_{b_{k+1}}^{b_k}} + \sum_{(l,j)\in C}\left\|r_C\left(\hat{z}_l^{c_j},\chi\right)\right\|^2_{P_l^{c_j}}\right\} \tag{20}$$

where $\{r_p, H_p\}$ is the prior information from the marginalization operation, which encodes marginalized information (see the states inside the dashed rectangle of Figure 4). The variable $r_{\mathcal{B}}(.)$ is the residual term for IMU pre-integration (see Equation (13)). The variable $r_C(.)$ is the residual term for visual re-projection (see Equation (19)). $\mathcal{B}$ denotes the set of all IMU measurements and $C$ is the set of features that were observed at least twice in the current sliding window. $P_{b_{k+1}}^{b_k}$ denotes the information matrix for IMU pre-integration. $P_l^{c_j}$ denotes the information matrix for visual re-projection, which represents the uncertainty of feature measurements. In [6], $P_l^{c_j}$ is fixed and is correlated with the focal length. The information matrix is the inverse of the covariance matrix. The fixed information matrix can work well in an ideal scenario. Unfortunately, the positioning result will be significantly misled by unmodeled outliers. Therefore, in the next section, we propose an online sensor model to adaptively model the uncertainty of visual measurements.

## 4. Online Sensor Model Estimation

According to our previous work in [11], the result shows that the dynamic feature points from dynamic objects are one of the major factors which degrade the performance of the VINS in the urban areas. In addition, the positioning error is highly correlated with the quality of feature tracking. We propose to mitigate the effects of dynamic objects by adaptively estimating the uncertainty of visual measurements based on the quality of feature tracking from two aspects, the adaptive covariance estimation and the adaptive M-estimator in the remainder of this section.

*4.1. Adaptive Covariance Estimation*

Based on our findings in [11], we propose to correlate the uncertainty of a given visual measurement with two parts: (1) the quality of feature tracking which is determined by the number of times that the feature is tracked (NTFT). The more times we see the same feature, the better is the feature quality; (2) the geometry distribution factor ($g_{f,t}$) which is determined by the geometry of feature distribution (GFD). Assuming that a set of tracked features at a given epoch $t$ from the $j$th image are denoted by $F_t^j$ as follows:

$$F_t^j = \{f_{t,1}^j, f_{t,2}^j, \dots, f_{t,m}^j\} \tag{21}$$

where $m$ represents the number of features in the $j$th image. Each feature $f_{t,l}^j$ is represented by $f_{t,l}^j = \{u_{t,l}, v_{t,l}, N_{f,t,l}, \}$. $u_{t,l}$ and $v_{t,l}$ denote the pixel position of the feature in the image. $N_{f,t,l}$ denotes the number of times that feature $l$ is tracked.

Each feature corresponds to a landmark that constrains the pose of the camera in the VINS. Different geometry distribution of features can result in the different performance of state estimation of the system. The ideal condition is that all tracked features are uniformly distributed surrounding the center of the camera. Unfortunately, this is usually not available due to the complex environmental conditions in urban canyons. In other words, the distribution of features relies strongly on the distributions of surrounding objects, such as buildings, vehicles and others. Figure 6 shows the geometry distribution of the features in two different cases. Figure 6a shows a decent geometry distribution of features where the features distribute over the whole image. Figure 6b shows a case where a majority of the features located in the middle of the figure.

As shown in Figure 2, feature-based VINS positioning is similar to satellite-based GNSS positioning. The precision of GNSS positioning is highly related to the geometry distribution of satellites concerning the position of the GNSS receiver. The quality of the distribution is described using 3D position dilution of precision (PDOP) [46]. Inspired by this fact, we adopt a similar idea from GNSS positioning to describe the quality of geometry distribution of features, $g_{f,t}$.

Firstly, with the given estimated initial guess about the position of the camera and the detected features using standard state initialization [51], the observation matrix correlating the positions of both the camera and the 3D position of feature is derived as follows:

$$H = \begin{bmatrix} (x_1 - x)\lambda_1 & (y_1 - y)\lambda_1 & (z_1 - z)\lambda_1 \\ (x_2 - x)\lambda_2 & (y_2 - y)\lambda_2 & (z_2 - z)\lambda_2 \\ \vdots & \vdots & \vdots \\ (x_m - x)\lambda_m & (y_m - y)\lambda_m & (z_m - z)\lambda_m \end{bmatrix}_{m \times 3} \tag{22}$$

where $(x, y, z)$ denotes the position of the camera which can be derived from the system state $(P_{b_k}^w)$ at epoch $t$ and $(x_m, y_m, z_m)$ denotes the 3D position of the $m$th feature during epoch $t$. Based on the derivation of PDOP, the $Q$ matrix can be derived as follows:

$$Q = (H^T H)^{-1} \tag{23}$$

where $Q$ is a 3x3 matrix as follows:

$$Q = \begin{bmatrix} \sigma_x^2 & \sigma_{xy} & \sigma_{xz} \\ \sigma_{xy} & \sigma_y^2 & \sigma_{yz} \\ \sigma_{xz} & \sigma_{yz} & \sigma_z^2 \end{bmatrix} \tag{24}$$

where the $\sigma_x^2$, $\sigma_y^2$ and $\sigma_z^2$ denote the uncertainty associated with the geometry distribution. Smaller $g_{f,t}$ means that the features are more decentralized which can lead to better VINS estimation and vice versa. Therefore, $g_{f,t}$ is calculated as follows:

$$g_{f,t} = \sqrt{\sigma_x^2 + \sigma_y^2 + \sigma_z^2} \tag{25}$$

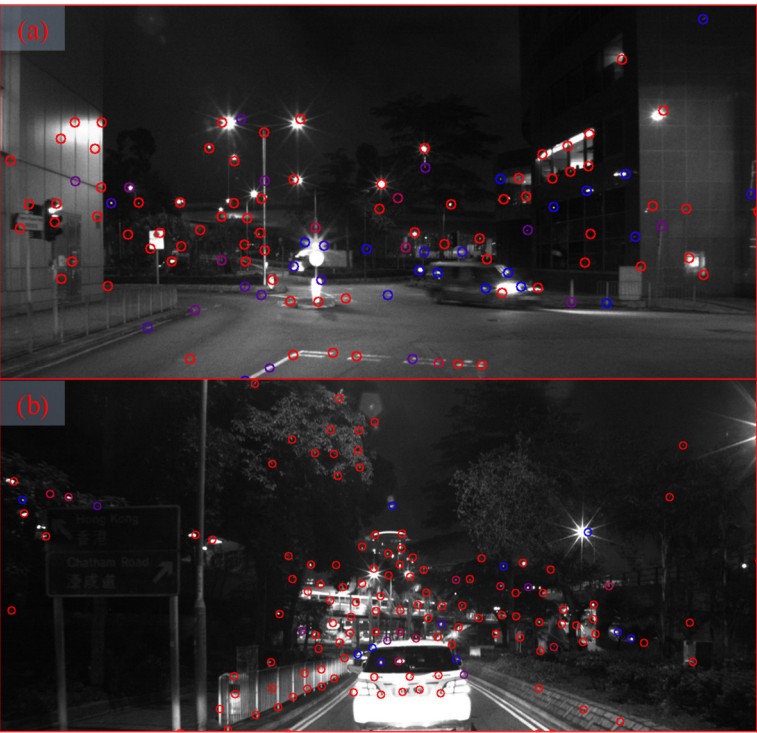

**Figure 6.** Illustration of the (**a**) decentralized feature distribution and (**b**) partially centralized feature distribution due to the environmental conditions. Circles represent the detected and tracked features. The red circles are features tracked by more times than the blue ones.

Therefore, the adaptive information matrix $(\Sigma_l^{c_j})$ is derived as follows:

$$\Sigma_l^{c_j} = P_l^{c_j} N_{f,t,l} \cdot \frac{1}{g_{f,t}} \cdot s \tag{26}$$

$$\text{with } P_l^{c_j} = \begin{bmatrix} \frac{F_c}{1.5} & 0 \\ 0 & \frac{F_c}{1.5} \end{bmatrix} \tag{27}$$

where $P_l^{c_j}$ represents the original information matrix from [6], the constant 1.5 denotes the standard deviation of pixels. The $F_c$ is the focal length of the virtual camera and the value is set to 460 in the framework [6]. The parameter of focal length describes the undistorted image. $s$ is the scaling factor which is experimentally determined and $F_c$ denotes the focal length of the given camera. Note that the covariance and the information matrix are mutually inversed.

## 4.2. Adaptive M-Estimator

The objective of the FGO is to minimize the summation of the residual function (20) to approach the optimal state set $\chi$. Unfortunately, the nonlinear function (20) is always a non-convex problem that has multiple sub-optimal, the local minimums phenomenon. The outlier measurements, which dominate the overall residual, can easily lead to local minimum estimation. Instead of de-weighting the outlier measurement by tuning the covariance matrix, the M-estimator [52] is a promising technique that enhances the resilience of the optimizer by using an additional robust function. However, the M-estimator relies heavily on parameter tuning. Figure 7 shows the state-of-the-art Huber-based M-estimator with different parameters based on (27). The curvature of the M-estimator relies heavily on different $k$ values, which is related to the robustness of M-estimator. The smaller variable k (black curve in Figure 7) could lead to a smoother curve of Huber function. As a result, the smoother curve can be more robust in mitigating the effects of outlier measurements. However, a $k$

with a too-small value can lead to an extremely small gradient of the error function (20), making it difficult for the optimizer to approach the optimal state. The research in [35,36] shows that extensive parameter tuning is required to obtain satisfactory performance using the M-estimator.

$$\rho(r) = \begin{cases} \dfrac{1}{2}r^2, |r| \le k \\ k\left(|r| - \dfrac{1}{2}k\right), \text{otherwise} \end{cases} \tag{28}$$

where $r$ denotes the residual measurement, $k$ denotes the parameter that needs to be tuned and $\rho(*)$ represents the robust Huber function.

Different from the offline tuned M-estimator [53] and the residual-based M-estimator [38] which relies on the initial guess about the state estimation, an adaptive M-estimator based on the Huber function by correlating the parameters of the M-estimator with the NTFT is proposed in this study, which could correct the visual residual model to further mitigate the effects of dynamic feature points. The parameter of the Huber function is estimated as follows at the given epoch $t$:

$$\rho\left(r_C(\widehat{\mathbf{Z}}_l^{c_j}, \boldsymbol{\chi}), N_{f,t,l}\right) = \begin{cases} \dfrac{1}{2}(r_C(\widehat{\mathbf{Z}}_l^{c_j}, \boldsymbol{\chi}))^2, |\, r_C(\hat{Z}_l^{c_j}, \boldsymbol{\chi})| \le k_t \\ k_t\left(|r_C(\widehat{\mathbf{Z}}_l^{c_j}, \boldsymbol{\chi})| - \dfrac{1}{2}k_t\right), \text{Otherwise} \end{cases} \tag{29}$$

$$k_t = M_s \ N_{f,t,l} \tag{30}$$

where $r_C(\widehat{\mathbf{Z}}_l^{c_j}, \boldsymbol{\chi})$ is the visual residual of feature $l$ in the $j$th image, $k_t$ is the parameter of the Huber function and $M_s$ is a scaling factor correlating $N_{f,t,l}$ and $k_t$ which is pre-determined. Therefore, an adaptive M-estimator is derived as (28).

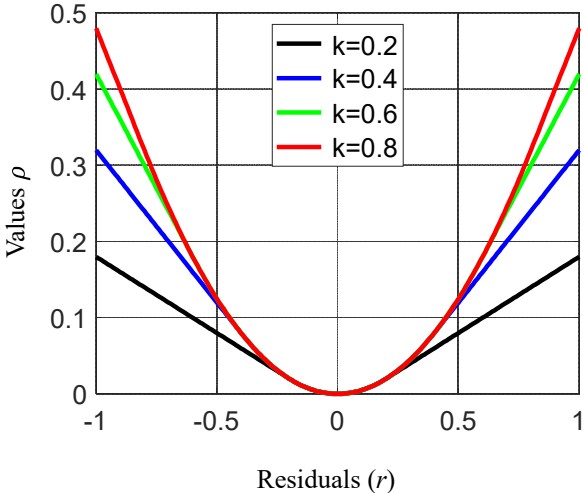

**Figure 7.** Huber M-estimator with different coefficients. The x-axis denotes the value of residual, corresponding to $r_C(\hat{Z}_l^{c_j}, \boldsymbol{\chi})$. The y-axis denotes the value of robust function given the residual.

### 4.3. Visual-Inertia Optimization with Online Sensor Model

Based on the VINS system derived in Section 3 and the online sensor model in sections 4.1 and 4.2, the new objective function is as follows:

$$\min_{\chi} \left\{ \|r_p - H_p\chi\|^2 + \sum_{k\in\mathcal{B}} \left\|r_{\mathcal{B}}\left(\hat{Z}_{b_{k+1}}^{b_k}, \chi\right)\right\|_{P_{b_{k+1}}^{b_k}}^2 + \sum_{(l,j)\in C} \left\|\rho\left(r_C(\hat{Z}_l^{c_j}, \chi), N_{f,t,l}\right)\right\|_{\Sigma_l^{c_j}}^2 \right\} \qquad (31)$$

Where $\rho(*)$ is the proposed adaptive M-estimator based on Huber function. The variable $\Sigma_l^{c_j}$ denotes the adaptive information matrix based on Equation (26). Similar to Equation (20), the $\{r_p, H_p\}$ denotes the prior information from the marginalization operation. $r_{\mathcal{B}}(.)$ is the residual term for IMU pre-integration. The $r_C(.)$ is the visual residual term. The detailed information of the residual terms is presented in Section 3.2 and 3.3. The major difference with the Equation (20) is the robust Huber function $\rho(*)$ and the adaptive information matrix $\Sigma_l^{c_j}$, which are derived in sections 4.1 and 4.2.

## 5. Experimental Results

### 5.1. Experimental Setup

To validate the effectiveness of the proposed method, two experiments were conducted in typical urban canyons of Hong Kong. The experiment setup is shown in the left side of Figure 8. An Xsens MTi 10 IMU was employed to collect raw IMU measurements at a frequency of 200 Hz. A monocular camera (BFLY-U3-23S6C-C) was employed to collect color images at a frequency of 10 Hz. In addition, the NovAtel SPAN-CPT, a GNSS (GPS, GLONASS and BeiDou) RTK/INS (with fiber-optic gyroscopes) integrated navigation system, was used to provide the ground truth of positioning. The gyro bias in-run stability of the FOG was 1 degree per hour and its random walk was 0.067 degree per hour. The baseline between the rover and the GNSS base station was about 7 km. All the data were collected and synchronized based on the time stamp provided by the robot operation system (ROS) [54]. The coordinate systems between all the sensors were calibrated before the experiments. Figure 8a,b shows the two evaluated urban canyons. The tested Urban Canyon 1 contains mainly static environmental structures and limited dynamic objects. The tested Urban Canyon 2 was significantly more challenging with numerous dynamic objects and complex environmental structures. More important, the dataset in Urban Canyon 2 was collected during the evening period with unstable and challenging illumination condition where was interesting to see how the proposed method can work.

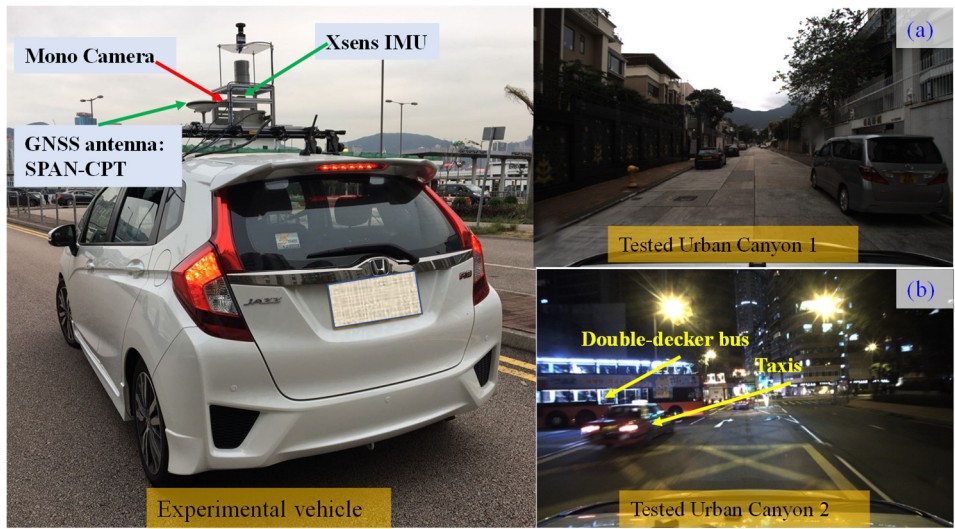

**Figure 8.** Sensor setup and scenes of the evaluated dataset.

To verify the performance of the proposed method, several methods were compared.
(1)  VINS: the original VINS solution from [6].

(2) VINS–AC: VINS-aided by adaptive covariance estimation proposed in Section 4.1.
   a. VINS-Adaptive covariance ($g_{f,t}$): only consider $g_{f,t}$ during covariance estimation
   b. VINS-Adaptive covariance ($g_{f,t}, N_{f,t,l}$): consider both $g_{f,t}$ and $N_{f,t,l}$ during covariance estimation.
(3) VINS–AC–ME: VINS-aided by adaptive covariance estimation proposed in Section 4.1 and adaptive M-estimator in Section 4.2.

Regarding the accuracy evaluation of the listed four methods above, we make use of the popular EVO toolkit [55], which was extensively used for the evaluation of SLAM algorithms. The parameters used in the experiments are shown in Table 1. The window size n was set to 11 frames in the optimization, which includes 10 keyframes and 1 newest frame. $F_c$ was the focal length and both of the $s$ and $M_s$ are the scaling factor.

**Table 1.** Parameter values used in this study.

| Parameters | Value | Parameters | Value |
|:---:|:---:|:---:|:---:|
| $F_c$ | 460 | $b_a$ | 0.001 m/s2 |
| $s$ | 0.02 | $b_\omega$ | 0.0001 rad/s |
| $M_s$ | 0.02 | $n_a$ | 0.1 m/s2 |
| $n$ | 11 | $n_\omega$ | 0.01 m/s |

*5.2. Evaluation of the Dataset Collected in Urban Canyon 1*

An experiment in Urban Canyon 1 was firstly conducted to validate the performance of the proposed method. The positioning results for the listed four methods are shown in Table 2. The mean error was defined by the RPE (relative pose error) in the EVO toolkit [55]. The mean error of the VINS was 0.33 meters, with the maximum error reaching 1.84 meters. Be noted that this was original the performance based on the work in [6]. With the geometry distribution of features ($g_{f,t}$) considered in the adaptive covariance, the positioning error decreases to 0.32 meters. The positioning error was slightly improved by 3.03%, which was calculated by the division of improvement and the error of original VINS. Furthermore, with the help of the number of feature-tracking times ($N_{f,t,l}$) in the adaptive covariance estimation, the mean error decreases to 0.30 meters, with an improvement of 9.09%. With the help of the adaptive M-estimator, the mean error stays at 0.30 meters, with the maximum error decreasing to 1.44 meters. The positioning error of the proposed method (VINS–AC–ME) was slightly mitigated by 9.09%. The fifth row shows the percentage of drift which is calculated by the formulation as follows:

$$\frac{|P_{vins} - P_{GT}|}{D}\%$$ (32)

where $P_{vins}$ denotes the pose estimation finally epoch from VINS and $P_{GT}$ denotes the pose estimation at the last epoch from the ground truth. D denotes the total driving distance. The percentage evaluates the accumulated drift of VINS at the final epoch. Different from the relative positioning error (RPE), the "% drift per distance" relies heavily on drift direction. As the following Figure 9 shows that the proposed method VINS–AC–ME drifts significantly, while the mean error decreases to 0.3 meters from 0.33 meters. The result shows that the proposed method can help to improve the performance of VINS even in the evaluated Urban Canyon 1 with limited dynamic objects.

**Table 2.** Positioning performance comparisons of listed methods in Urban Canyon 1.

| All Data | VINS [6] | VINS–AC ($g_{f,t}$) | VINS–AC ($g_{f,t}, N_{f,t,l}$) | VINS–AC–ME |
|---|---|---|---|---|
| Mean error | 0.33 m | 0.32 m | 0.30 m | 0.30 m |
| Std | 0.31 m | 0.30 m | 0.30 m | 0.29 m |
| Max error | 1.84 m | 1.35 m | 1.70 m | 1.44 m |
| % drift per meters | 2.16% | 1.85% | 2.69% | 2.71% |
| Improvement | - | 3.03% | 9.09% | 9.09% |
| Percentage of Outliers | 8.6% | 8.6% | 8.6% | 8.6% |

The trajectories of the listed methods and the reference trajectory are shown in Figure 9. The total length of the trajectory was 1186.081 meters. Overall, the trajectory of the VINS–AC–ME (see blue curve) was the one closest to the reference trajectory (see black curve). The relative positioning error throughout the test is shown in Figure 10. The accuracy of the proposed method was slightly improved with the help of the proposed online sensor model adaption. This was because the static feature points dominate the visual measurements in urban canyons. However, the proposed method can obtain slightly improved performance in the evaluated Urban Canyon 1.

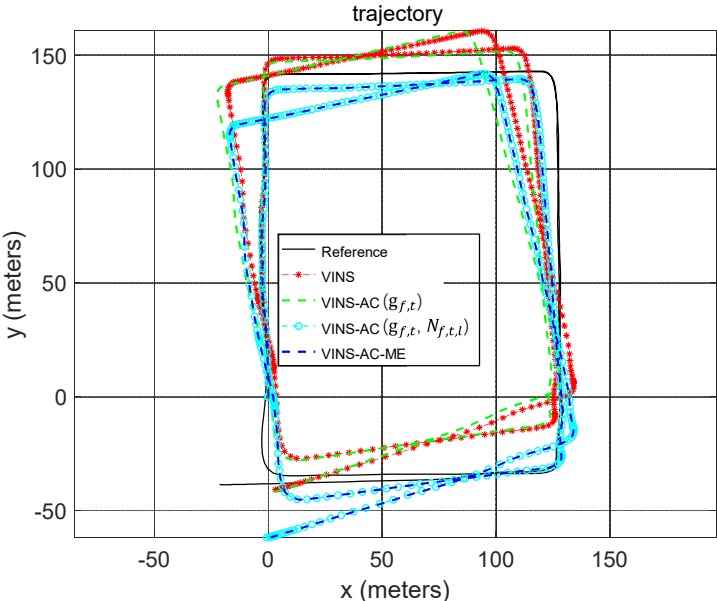

**Figure 9.** Trajectories of the VINS and proposed method and reference trajectory in Urban Canyon 1.

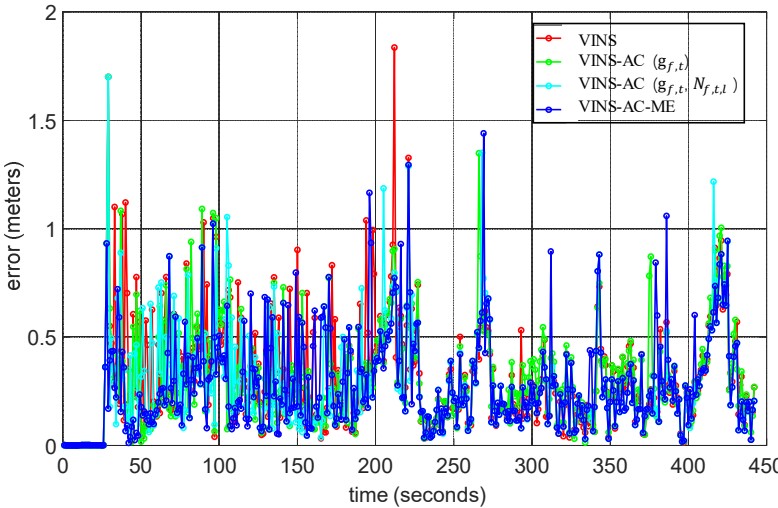

**Figure 10.** Relative positioning errors of tested methods in Urban Canyon 1.

The last row of Table 2 shows the percentage of outlier features, which was calculated by the division of the total number of outlier features and the total number of the tracked features throughout the evaluated dataset. During the calculation, we classify the outlier measurement based on the $N_{f,t,l}$. One feature was identified as an outlier measurement if the variable $N_{f,t,l}$ was less than 10. Be noted that this value was only used for evaluation purposes and was not used in the model adaption of visual measurements. Based on this rule, 8.6% of the features were outlier measurements.

### 5.3. Evaluation of the Data Collected in Urban Canyon 2

To challenge the performance of the proposed method, we conduct the other experiment in Urban Canyon 2 with numerous dynamic objects and the data were collected at night (see Figure 8b). In this experiment, the number of DFPs was far more than that in Urban Canyon 1, with a percentage of 19.6% which can be seen in Table 3. Therefore, we believe that Urban Canyon 2 was more challenging for our proposed online sensor model compared to the evaluated Urban Canyon 1.

The positioning results of the listed methods are shown in Table 3. The mean error of the original VINS was 0.79 meters, with the maximum error reaching 5.58 meters. The mean error decreases to 0.69 meters with the geometry distribution of features ($g_{f,t}$) considered in the adaptive covariance. The number of feature-tracking times ($N_{f,t,l}$) was also considered in the adaptive covariance and the mean error decreases to 0.64 meters, with an improvement of 18.99%. With the help of the adaptive M-estimator, the mean error decreases to 0.59 meters, with an improvement of 25.32%.

Similarly, the % drift per meter was also shown in the fifth row of Table 3, which was calculated based on (31). Interestingly, we can see that the value of "% drift per meter" was even larger after applying the proposed method (3.73%), compared with the original VINS (2.1%). This was mainly because the drift direction of the proposed method was significantly different from the original VINS, which can be seen in Figure 11. However, the proposed method provides more accurate relative positioning which can be seen by the mean error. Therefore, the VINS can provide accurate positioning relatively in a short period. Integration of the proposed VINS, which is subjected to drift, and the drift-free GNSS positioning is a promising solution to provide accurate and globally reference positioning.

**Table 3.** Positioning performance comparisons of listed methods in Urban Canyon 2.

| All Data | VINS [6] | VINS–AC ($g_{f,t}$) | VINS–AC ($g_{f,t}$, $N_{f,t,l}$) | VINS–AC–ME |
|---|---|---|---|---|
| Mean error | 0.79 m | 0.69 m | 0.64 m | 0.59 m |
| Std | 0.96 m | 0.86 m | 0.84 m | 0.75 m |
| Max error | 5.58 m | 6.39 m | 7.32 m | 7.26 m |
| % drift per meters | 2.1% | 2.04% | 4.10% | 3.73% |
| Improvement | - | 12.66% | 18.99% | 25.32% |
| Percentage of outliers | 19.6% | 19.6% | 19.6% | 19.6% |

Figure 11 shows the trajectories of the listed methods and the reference trajectory. The total length of the trajectory was about 1984.448 meters. We can see that the proposed method (blue curve) is the one closest to the reference trajectory (black curve). The details about the relative positioning error are shown in Figure 12.

To show the details about the improvement, four epochs are selected in Figure 12 and the snapshots of the selected epochs are shown in Figure 13. The corresponding positioning errors are shown in Table 4. We can see from Table 4 that the error of the VINS reaches the maximum value of 5.59 meters during epoch 260 (A). With the geometry of features distribution ($g_{f,t}$) considered in the adaptive covariance, the error of the VINS–AC ($g_{f,t}$) decreases to 2.81 meters. Moreover, the number of feature-tracking times ($N_{f,t,l}$) was also introduced to the adaptive covariance and the error of the VINS–AC ($g_{f,t}$, $N_{f,t,l}$) decreases to 1.62 meters, which shows that $g_{f,t}$ and $N_{f,t,l}$ can model the uncertainty of each feature measurement to improve the performance of the VINS. With the help of the adaptive M-estimator, the error of the proposed method (VINS–AC–ME) decreases to 1.02 meters. A similar condition appears during epoch 343 (B). The proposed method outperforms the VINS in positioning accuracy. Based on the proposed adaptive covariance (26), $g_{f,t}$ and $N_{f,t,l}$ were used to evaluate the uncertainty of the visual measurements and Figure 14 and Figure 15 show that the system tends to rely on the visual measurements during epoch 260 (A) and epoch 343 (B). However, we find out that the proposed method leads to a large positioning error during epoch 29 (C). The error of the VINS–AC ($g_{f,t}$) increases to 6.39 meters and the error of the VINS–AC ($g_{f,t}$, $N_{f,t,l}$) even increases to 7.26 meters. The error of the VINS–AC–ME was also 7.26 meters. This was due to the fact that the proposed system tends to assign a higher weight to the visual measurement, while the quality of the tracked features was poor during the epoch.

As can be seen from Figure 13c, the feature tracked on a blurred image. Therefore, the positioning error increases significantly during epoch 29 (C). Interestingly, the error of the VINS–AC ($g_{f,t}$, $N_{f,t,l}$) can reach 7.32 meters during epoch 127 (D), which was far larger than that of the VINS–AC ($g_{f,t}$) (1.43 meters). The error was mainly caused by other factors, such as the unstable illumination conditions. With the help of the adaptive M-estimator, the error of the VINS–AC–ME decreases to 1.32 meters, which shows that the adaptive M-estimator can enhance the resistance to outliers.

Figure 16 shows that the adaptive M-estimator can correct the visual residual model by using an additional robust function, especially in challenging urban canyons. The red curve denotes the visual residual on the VINS–AC ($g_{f,t}$, $N_{f,t,l}$) and the blue curve denotes the residual on the VINS–AC–ME. The major difference between the two methods was whether the M-estimator was utilized. Overall, we can see from Figure 16 that the residual with large values (see red curve) was significantly mitigated.

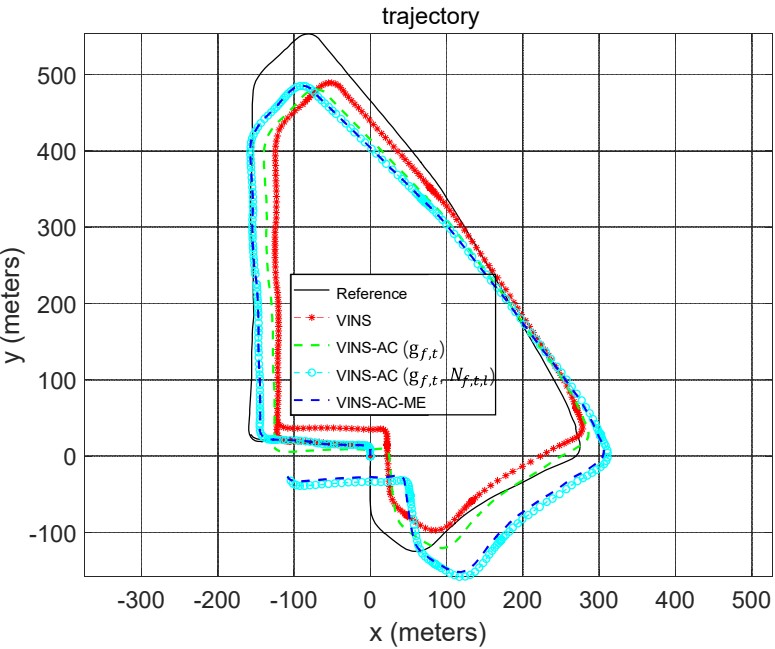

**Figure 11.** Trajectories of the VINS and proposed method and reference trajectory in Urban Canyon 2.

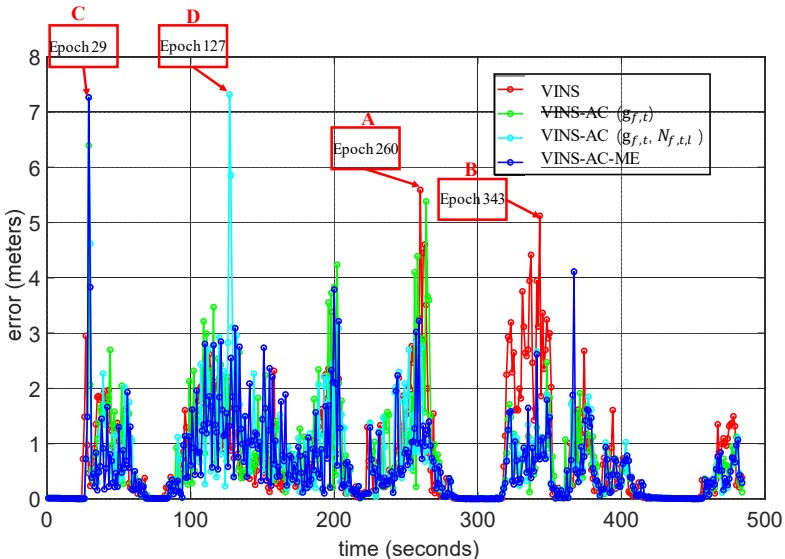

**Figure 12.** Relative positioning errors of tested methods in Urban Canyon 2.

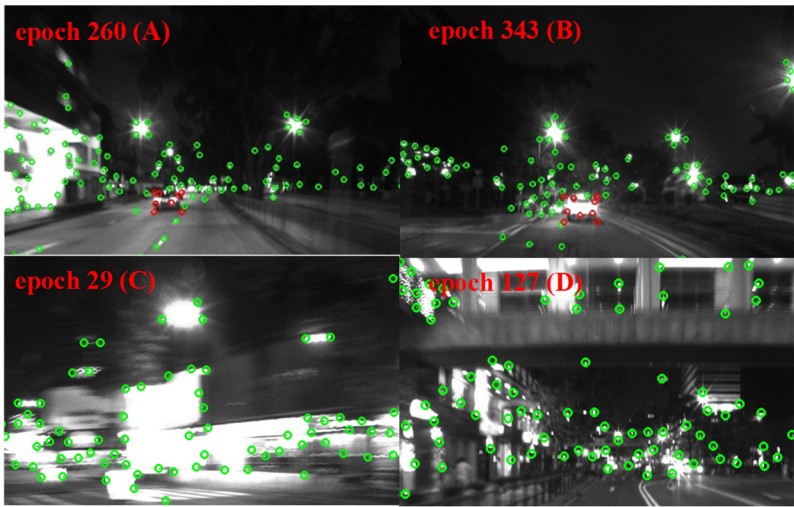

**Figure 13.** Images of tested Urban Canyon 2 in the four selected epochs concerning Figure 12. Green circles denote the static feature points and the red circles denote the dynamic feature points.

**Table 4.** Positioning performance comparison between listed methods on the four selected epochs in Figure 12.

| Mean Error | VINS | VINS–AC ($g_{f,t}$) | VINS–AC ($g_{f,t}$, $N_{f,t,l}$) | VINS–AC–ME |
|---|---|---|---|---|
| Epoch 260 (A) | 5.59 m | 2.81 m | 1.62 m | 1.02 m |
| Epoch 343 (B) | 5.12 m | 0.98 m | 0.78 m | 0.72 m |
| Epoch 29 (C) | 0.47 m | 6.39 m | 7.26 m | 7.26 m |
| Epoch 127 (D) | 1.65 m | 1.43 m | 7.32 m | 1.32 m |

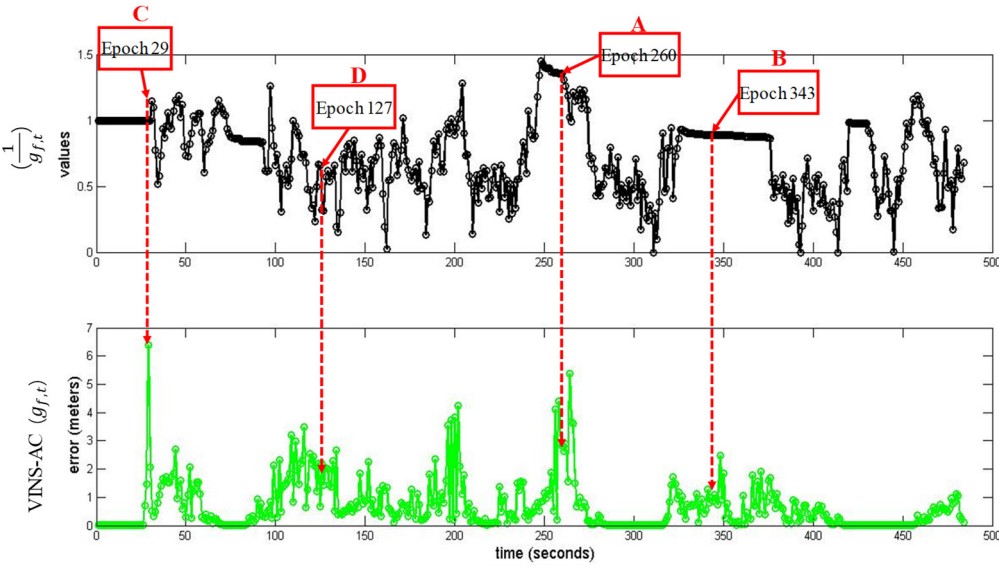

**Figure 14.** (top) Geometry of feature distribution ($g_{f,t}$) on the error of the VINS–AC ($g_{f,t}$); (bottom) error of the VINS–AC ($g_{f,t}$).

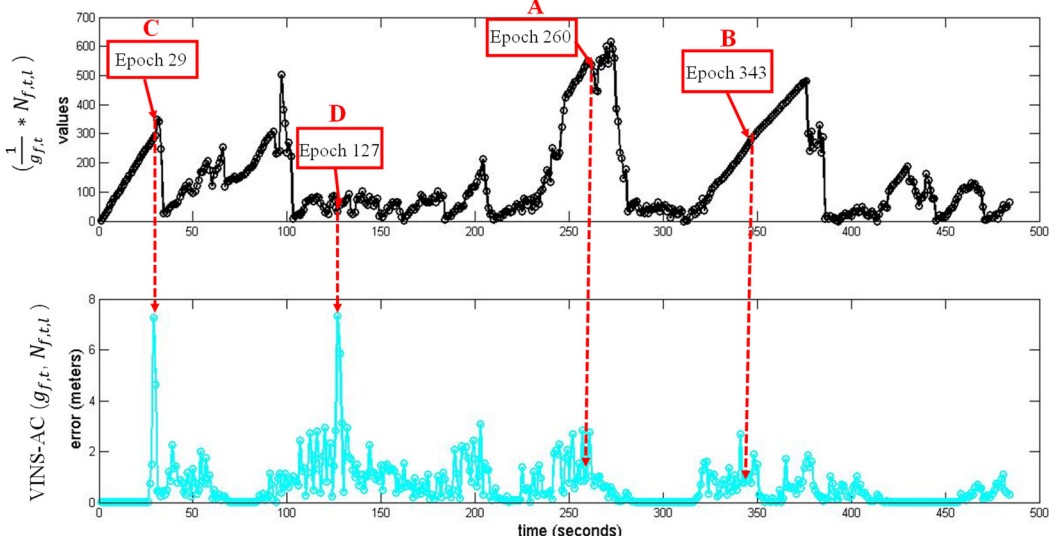

**Figure 15.** (top) Geometry of feature distribution ($g_{f,t}$) and the number of feature tracking times ($N_{f,t,l}$) on the error of the VINS–AC ($g_{f,t}$ , $N_{f,t,l}$); (bottom) error of the VINS–AC ($g_{f,t}$, $N_{f,t,l}$).

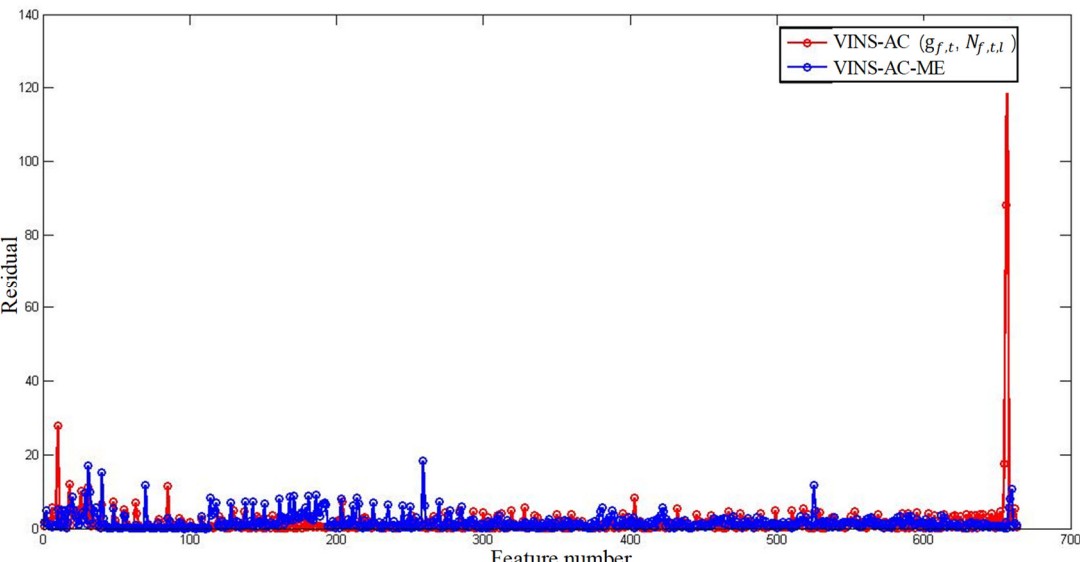

**Figure 16.** Residual comparison between VINS–AC ($g_{f,t}$, $N_{f,t,l}$) and VINS–AC–ME. Red curve denotes the visual residual on the VINS–AC ($g_{f,t}$, $N_{f,t,l}$) and the blue curve denotes the residual on the VINS–AC–ME.

## 6. Conclusions and Future Work

Achieving accurate positioning via VINS in an urban canyon is a challenging problem, due to the numerous expected dynamic objects. Instead of directly eliminating the features from dynamic objects, this study proposes to adopt the visual measurement model based on the quality of feature tracking to improve the performance of the VINS. To model the uncertainty of visual measurements and improve the system's resistance to outliers, adopting the adaptive covariance and the adaptive M-estimator to evaluate the performance of the VINS is proposed in this study. The accuracy is improved in both of the two experiments, especially in Urban Canyon 2, which shows the effectiveness of the proposed method.

This study only contributes to mitigating the effects of DFPs. The remaining errors were mainly caused by other factors, such as the unstable illumination conditions and feature extraction failure.

In the future, we will further study how to acquire and estimate the quality of the detected features for VINS and its integration with other sensors (e.g., GNSS) in urban canyons.

**Author Contributions:** Conceptualization, X.W.B. and L.-T.H.; methodology, X.W.B.; software, X.W.B.; formal analysis, X.W.B. and W.S.W.; data collection, X.W.B. and W.S.W.; writing—original draft preparation, X.W.B.; writing—review and editing, X.W.B., W.S.W. and L.-T.H.; supervision, L.-T.H. All authors have read and agreed to the published version of the manuscript.

**Funding:** This research was funded by The Hong Kong Polytechnic University. The project ZVKZ was "Positioning and Navigation for Autonomous Driving Vehicle by Sensor Integration".

**Conflicts of Interest:** The authors declare no conflict of interest.

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
