# Peer review of "Robust Visual-Inertial Integrated Navigation System Aided by Online Sensor Model Adaption for Autonomous Ground Vehicles in Urban Areas"

_remotesensing, doi:10.3390/rs12101686_

Round 1

Reviewer 1 Report

The authors tried to improve the accuracy of VINS in a dynamic environment through a self-tuning covariance and an adaptive M-estimator. There are merits to the approach and showed to produce improved results in experiments.

There are some minor comments on the paper.

  1. The sentence structure in line 422 to 423 "The smaller is k ... it is meanwhile" needs to be improved.
  2. In Figure 7, I am not sure what is the x axis, it is good to label it. If the x axis is r, then the quadratic part of the function is not properly shown.
  3. It is good to show the error of the experiment in terms of % drift per distance traveled.
  4. The positioning error is shown as a % but did not indicate it is a percentage of what parameter.
  5. In the trajectory plots in Figure 9 and 11, it is better to use solid, dashed, dotted lines in addition to colored lines.
  6. It is good to show the percentage of DFP as compared to the total features points. This will highlight the difference between the two experimental datasets.

Author Response

Dear Review,

Thanks for your kind comments. The paper is revised to include all the suggested revisions. Point-by-point responses to the comments are provided in the attached file. All the revised parts were highlighted with the red color and “Track Changes” in Microsoft Word.

Reviewer 2 Report

This paper advocates the use of a visual measurement model to improve the performance of VINS (Visual-Inertial Integrated Navigation System), based on the quality of feature tracking. The proposed approach adopts a self-tuning covariance estimation method to model the uncertainty of each feature measurements and an adaptive M-estimator to correct the measurement residual model to further mitigate the impacts of outliers.

Generally the paper is well structured and the various parts well developed.
The introduction gives a general overview of the problem and contains the related works, correctly contextualizing the work presented.
The method is accurately described in the technical part, and the choices are justified with sufficient detail.
The experiments are exhaustive even if limited in number, taking into account that some of the main problems of this type of systems, such as the quality of images acquired in a poorly lit urban environment, are outside the scope of this work.

My main criticism goes instead to the linguistic exposure, which must necessarily be improved in all parts of the paper. Many sentences in the paper are grammatically incorrect or incomprehensible except after a careful re-reading of the whole paper.

In order for the paper to be published an extensive editing of language and style is required.

Author Response

Dear Review,

Thanks for your kind comments. To improve the language quality of the manuscript, we have made the manuscript proofed and edited by a professional paper editing institute. The certificate of paper editing is attached to the attached response letter. The revisions are highlighted using red color and the “Track Changes” in Microsoft Word of the revised manuscript.

Reviewer 3 Report

The paper titled: "Robust Visual-Inertial Integrated Navigation System Aided by Online Sensor Model Adaption for Autonomous Ground Vehicles in Urban Areas", is presenting a genegic method for performing visual navigation in dynamic scenarios (with many moving objects). The suggested method is based on approximating the quality of feature tracking.

The problem of visual navigation and mobile mapping in dynamic urban regions is both very important and challenging.

The paper has few major problems in its current state:

  1. The "related papers" section should mention related projects such as:https://www.sciencedirect.com/science/article/pii/S1574013716301794, but also software tools such as: AR-Core (see:https://developers.google.com/ar), AR-Kit, and some other papers addressing the detection of moving objects (not detection of objects which might be moving - aka cars). 
  2.  Recent results of papers such as: https://www.sciencedirect.com/science/article/pii/S1574013716301794, https://ieeexplore.ieee.org/abstract/document/8662921 allow a realtime detection of moving objects, naturally one can apply such an algorithm and simply ignore feature points that "falls" within such moving objects. This way feature points on "parking cars" will be used for ego-motion (or visual navigation).
  3. There are few publicly available benchmarks such as: http://www.cvlibs.net/datasets/kitti/ and several available software for visual navigations tools (e.g., AR-Core, SVO2.0, etc') - The authors should compare their results to at least one other leading solution. Alternatively (yet less recommended) the authors can publish their benchmark in a way that visual navigation tools will be able to run it.
  4. The results presented (e.g., figures 9,11) seem to have a significant drift or error (with respect to the GT reference). The SVO2.0 implementation seems to have significantly better results (on related dynamic scenarios).

Author Response

Dear Review,

Thanks for your kind comments. The paper is revised to include all the suggested revisions. Point-by-point responses to the comments are provided in the attached file. All the revised parts were highlighted with the red color and “Track Changes” in Microsoft Word of the revised manuscript.

Reviewer 4 Report

  1. Some qualitative explanation should be presented in the manuscript.
  2. One of the main idea of the manuscript seems to be eq. (30). The procedure of the optimization would be shown in more detail.
  3. Design parameter of the proposed algorithm should be presented and what values were set in the experiments.
  4. How many outliers were detected in the experimental scenarios and the results should be presented. The results should be compared with that in [5].
  5. It would be better to revise English in the manuscript.
  6. Some terms should be changed. Please check all over the manuscript.
  7. Please check figures.   

Author Response

Dear Review,

Thanks for your kind comments. The paper is revised to include all the suggested revisions. Point-by-point responses to the comments are provided in the attached file. All the revised parts were highlighted with the red color and “Track Changes” in Microsoft Word of the revised manuscript.

To improve the language quality of the manuscript, we have made the manuscript proofed and edited by a professional paper editing institute. The certificate of the paper editing is also attached to the response letter.

Round 2

Reviewer 3 Report

The paper's writing (overall structure) was improved significantly.

Having an open-source benchmark for visual navigation is surely a good thing.

That said, my main concern remains: It is not clear to me what is the suggested improvement over existing solutions. I Strongly recommend comparing the suggested method with leading visual navigation solutions and clearly state the "diff" and the potential novelty and improvements of the suggested method.

You might want to compare your method to existing ROS based Visual navigation (VSLAM / RGBD-SLAM) solutions - some of them can even be found in commercial devices such as Intel T265 tracking camera - which I have testes in dynamic scenarios (mostly walking in crowded regions) - and overall, it seems to allow a lower drift than your system - with strong loop closure mechanism. You might want to explain how your system can be beneficial for such devices.

Author Response

Dear Reviewer,

Thanks for your kind comments. We make a performance analysis of MSCKF using our dataset collected in urban canyons of Hong Kong. Please see the attachment. 

Best regards,

Reviewer 4 Report

The revised version of the manuscript was well-written.

Author Response

Dear Review,

Thanks for all your review comments.

Best Regards,
